# Transcription of ncRNAs promotes repair of UV induced DNA lesions in *Saccharomyces cerevisiae* subtelomeres

**Laetitia Guintini**[1⊙], **Audrey Paillé**[1⊙], **Marco Graf**[2], **Brian Luke**[3], **Raymund J. Wellinger**[1]*, **Antonio Conconi**[1]*

**1** Department of Microbiology and Infectious Diseases at the Université de Sherbrooke, Sherbrooke, Canada, **2** Institute for Developmental and Neurobiology (IDN) at the Johannes-Gutenberg-University, Mainz, Germany, **3** Institute of Molecular Biology (IMB), Mainz, Germany

⊙ These authors contributed equally to this work.
* Raymund.Wellinger@usherbrooke.ca (RJW); Antonio.Conconi@Usherbrooke.ca (AC)

**Data Availability Statement:** All raw files are available from the Mendeley database here https://data.mendeley.com//datasets/j9869tptct (DOI:10.17632/j9869tptct).

## Abstract

Ultraviolet light causes DNA lesions that are removed by nucleotide excision repair (NER). The efficiency of NER is conditional to transcription and chromatin structure. UV induced photoproducts are repaired faster in the gene transcribed strands than in the non-transcribed strands or in transcriptionally inactive regions of the genome. This specificity of NER is known as transcription-coupled repair (TCR). The discovery of pervasive non-coding RNA transcription (ncRNA) advocates for ubiquitous contribution of TCR to the repair of UV photoproducts, beyond the repair of active gene-transcribed strands. Chromatin rules transcription, and telomeres form a complex structure of proteins that silences nearby engineered ectopic genes. The essential protective function of telomeres also includes preventing unwanted repair of double-strand breaks. Thus, telomeres were thought to be transcriptionally inert, but more recently, ncRNA transcription was found to initiate in subtelomeric regions. On the other hand, induced DNA lesions like the UV photoproducts must be recognized and repaired also at the ends of chromosomes. In this study, repair of UV induced DNA lesions was analyzed in the subtelomeric regions of budding yeast. The T4-endonuclease V nicking-activity at cyclobutene pyrimidine dimer (CPD) sites was exploited to monitor CPD formation and repair. The presence of two photoproducts, CPDs and pyrimidine (6,4)-pyrimidones (6-4PPs), was verified by the effective and precise blockage of Taq DNA polymerase at these sites. The results indicate that UV photoproducts in silenced heterochromatin are slowly repaired, but that ncRNA transcription enhances NER throughout one subtelomeric element, called Y', and in distinct short segments of the second, more conserved element, called X. Therefore, ncRNA-transcription dependent TCR assists global genome repair to remove CPDs and 6-4PPs from subtelomeric DNA.

## Author summary

Our skin is constantly exposed to sunlight and the ultraviolet component of it can severely damage the DNA of our chromosomes. If that damage is not efficiently repaired, the cells'

**Funding:** This work was supported by funds received from the Natural Sciences and Engineering Research Council of Canada (NSERC) (to A.C.); the Canadian Institutes of Health Research (CIHR; FDN154315), as well as funds from the Canadian Research Chair on Telomere Biology and the Center for Research on Aging (CdRV) to R.J.W. Work in the lab of BL was supported by the Heisenberg program of the Deutsche Forschungsgemeinschaft (DFG, German Research Foundation, grant number LU 1709-2-1). The funders had no role in study design, data collection and analysis, decision to publish, or preparation of the manuscript.

**Competing interests:** The authors have declared that no competing interests exist.

physiology becomes deregulated and very often cancer ensues. The specific molecular mechanism that will remove this damage is called nucleotide excision repair or NER. NER is conserved from humans to yeast, and it is much more efficient on DNA that is transcribed into RNA. Here we report how NER acts at the very ends of the chromosomes, the telomeres. In particular, the results show that in this area of the chromosomes with very few genes and where transcription is kept very low, the remaining transcription of non-coding RNAs such as TERRAs still stimulates NER and therefore helps guarding the integrity of DNA. These findings therefore suggest that the spurious transcription of subtelomeric DNA has a very positive impact on DNA repair efficiency. Hence, in addition to the known functions of TERRA and other ncRNAs in telomere maintenance, their transcription *per se* can be viewed as a genome stabilizing function.

## Introduction

Living organisms are constantly exposed to DNA damaging agents that are present in the environment, like the ultraviolet (UV) spectrum of sun light. Primarily UVC ($< 280$ nm) and UVB (280–315 nm) radiations cause the formation of covalent bonds between adjacent pyrimidines. The resulting cyclobutene pyrimidine dimers (CPDs) and pyrimidine-6,4-pyrimidone photo-products (6,4-PPs) are highly cytotoxic as they block DNA replication and transcription [1]. Cells respond to UV induced DNA damage with the arrest of cell cycle and the activation of nucleotide excision repair (NER). One of the two NER sub-pathways is global genome NER (GGR) that removes DNA damage from silent heterochromatin, inactive chromatin and the non-transcribed strand (NTS) of transcriptionally active genes. The other sub-pathway is transcription coupled NER (TCR) that removes DNA damage from the transcribed strand (TS) of active genes [1]. GGR and TCR rely on different proteins for the recognition of DNA damage, but they share the same NER factors for incision of the damaged DNA strand and excision of the DNA lesion. TCR is triggered by stalled RNA polymerases at DNA lesions and removes DNA damage that blocks RNA polymerase elongation, re-establishing gene transcription. Likely for this reason, TCR is more efficient than GGR, and pyrimidine dimers (PDs) are removed faster from the TS than from the NTS of active genes [2–5]. In *Saccharomyces cerevisiae*, here referred to as yeast, TCR relies on the protein Rad26, a DNA dependent ATPase of the SWI2/SNF2 chromatin-remodeling family. Consequently, in the Rad26 depleted strain (*rad26Δ*) TCR is inefficient [6,7].

Like replication and transcription, NER is challenged by the organization of DNA in chromatin [8]. The basic unit of chromatin—the nucleosome—modulates the efficiency of NER and CPDs are repaired faster in linker DNA between nucleosomes or in DNA at the nucleosome edges, than in the center of nucleosomes [9–12]. Histone modifications, histone variants and chromatin remodeling are all essential to facilitate DNA damage recognition and repair in euchromatic nucleosomes [13,14]. Currently, less is known about the efficacy of NER and its requirements to recognize and repair pyrimidine dimers in heterochromatin [15]. In contrast to low condensed euchromatin that forms on transcribed regions of the genome, heterochromatin retains a condensed state throughout the interphase. In higher eukaryotes, heterochromatin can be in a facultative- or a constitutive- structure. The former depends on the developmental stage of cells, the latter forms on highly repetitive DNA sequences like pericentromeres, centromeres and telomeres [16]. In yeast, heterochromatin-like regions at and near the telomeres are characterized by the presence of silent information-regulators (Sir2, Sir3 and Sir4) that mediate histone tail de-acetylation and possibly the compaction of chromatin.

Remarkably, ectopic genes engineered close to yeast telomeres are silenced [17], a mark that was exploited to investigate NER in silenced genes and thought to be in a heterochromatin-like structure. It was reported that CPDs were inefficiently removed from the *URA3* gene inserted about two kilobases from a telomere [18,19]. In the absence of Sir2 or Sir3, the *URA3* gene was transcribed and CPDs were efficiently repaired. Hence, it was concluded that the efficiency of NER was reduced in the heterochromatin-like structure of synthetic subtelomeres [18,19]. However, natural telomeres are formed by terminal telomeric DNA repeats and the telomere associated Y'- and X- DNA elements, the former being present on about half of the telomeres [20]. On the distal telomeric DNA repeats there are no nucleosomes nor detectable histones [21,22]. Instead, these repeats are bound by Rap1 and a number of associated proteins, including Sir 2, 3 and 4 [20]. Nucleosomes and transcription factors, but no Sir proteins, are present on the Y'-element. Through the X-sequences, nucleosomes, Sir proteins and general transcription factors appear to form a heterochromatin-like structure [23,24]. Only recently we gained some information on the repair of pyrimidine dimers on natural yeast telomeres. Yeast need 4 hours to remove about 90–95% of CPDs from bulk genomic DNA [25], and the removal of CPDs from the Y'-element has similar characteristics (see below). Albeit slowly, yeast cells repair about 44% of photoproducts in the silenced X-element [12]. These results were unexpected because the nucleoprotein complexes at telomeres prevent repair pathways like homologous recombination and non-homologous end joining [26–28].

As introduced above, transcribing RNA polymerases promote the repair of UV induced DNA lesions in gene coding regions [4,6,25,29]. This characteristic of NER has received particular attention after the discovery of pervasive transcription in eukaryotic cells. High-throughput technologies revealed that approximately 85% of the yeast genome is transcribed. Among the transcripts are several noncoding RNAs (ncRNA), which are grouped into families that are categorized by their structures and functions, like the long noncoding RNAs (lncRNA) that stretch to more than 200 nucleotides and participate in cell regulatory networks [30–35]. For example, in contrast to the longstanding belief that the heterochromatic chromosome-ends are silenced, studies have shown that telomeres are transcribed into lncRNAs such as the housekeeping telomeric repeat-containing RNAs (TERRA) that are integral components of telomeric chromatin. Yeast TERRAs are transcribed by RNA polymerase-II in the 5' to 3' end-direction towards the telomere. They play important roles in processes like chromatin remodeling and regulation of telomerase activity [36]. In addition, TERRA reverse-strand transcripts were discovered for the Y'-telomeres and referred to as Sub-TERRA-cuts [36,37].

Thus, given that RNA polymerase-II is found on subtelomeres [36,38], TCR could occur as well but very little is known about TCR at the end of chromosomes. In this study, we investigated if ncRNA transcription fostered repair of pyrimidine dimers in Y'-elements, distal and adjacent to telomeric repeats, or in the X-element heterochromatin-like structure, with or without the Sir2 protein. The results show that Rad26 dependent-TCR occurred in both DNA strands of Y'-elements and in distinct, short segments of the X-element C-rich DNA strand, promoting repair of UV induced DNA damage in heterochromatin.

## Results

### Distinctive DNA strand-repair efficiency of CPDs in Y' sub-telomeres

Yeast telomeres are of two types, one has 1 to 4 copies of Y'-elements between the X-element and the terminal repeats, the other has only the X-element that borders the terminal telomeric repeats. In most strains, Y'-elements are found on 50 to 70% of all telomeres [20] (**Fig 1**). A recent study of the chromatin organization at the transition zone between subtelomeric sequences and the telomeric repeats revealed a surprisingly similar architecture for both types

of telomeres, with positioned nucleosomes and an array of basic transcription factors [39]. The sequence of the terminal repeats is ordered in a G-rich strand that runs 5' to 3' towards the end of the chromosome, and in a C-rich strand that runs 5' to 3' towards the centromere. Typically, the G-rich strand is 8–14 nts longer than the C-rich strand [20] (**Fig 1**). [Note: X- and Y'-element DNA sequences do not show noticeable base bias, but for practical reason we kept the 'G-rich' and 'C-rich strand' denomination for those regions as well].

Given the high homology of Y' element sequences, repair of UV induced DNA damage could only be followed within or at the 3'-end of undistinguished subtelomeres. Consequently, the results represented the average repair of all Y'-elements in the cell. To investigate NER in these regions, wild type (WT) yeast cells were grown to early log phase, UV irradiated and allowed to repair the DNA for various lengths of time. DNA was isolated from non-irradiated and irradiated cells and then digested with restriction enzymes. Because there are two forms of Y'-elements (6.7 and 5.2 kb), *HindIII* and *EcoRI* double digestion released 3 junction fragments of 4.4, 4.7 and 5.0 kb, as expected from permutations of two different adjacent Y'-elements (**Figs 1A** and **S1**). We considered the 3 fragments as one cluster with an average length of ~4.7 kb. The *HindIII* digestion released a 3.0 kb terminal fragment covering the telomere, and *Xho*I released a terminal fragment of 1.3 kb (**Fig 1A**). Because of the variable length of telomeric repeats, the two terminal fragments run as broad smeary bands in our gel-electrophoresis conditions. The enzyme T4 endo-V that nicks DNA at CPD sites was employed to determine the presence of CPDs. After denaturing agarose gel-electrophoresis and blotting, the filter

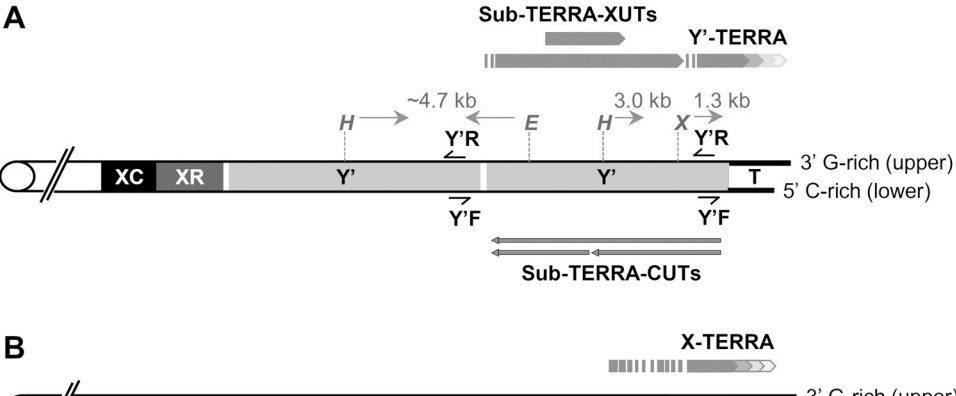

**Fig 1. Maps of ncRNA transcription in Y'- and X-element telomeres.** (A) Schema of a two copies Y'-element. Y'-elements (Y') are located between the X-element, that is formed by the XC and XR regions, and the terminal repeats (T). Y'-elements can be of 2 types, long (6.7 kb) and short (5.2 kb). The terminal repeat consists of 325 ±50 bp of imperfectly repetitive sequences ($C_{1-3}A/TG_{1-3}$), ending in a 3' single stranded overhang of 8 to 14 nts [61,62]. *HindIII* (H), *EcoRI* (E) and *XhoI* (X) denote the restriction enzyme-digestion sites, Y'F and Y'R denote the single-stranded oligonucleotides that were used as probes (**S2 Table**). Approximative chart and direction of ncRNA expression is indicated by pointy bars, with the widths that emphasize ncRNA abundance. TERRA are G-rich transcripts of 0.1 to 1.2 kb that begin at sub-telomere embedded promoters, they include sub-telomeric sequences and can extend to two thirds of the $TG_{1-3}$ repeats [36]. SubTERRA-XUTs are transcribed toward the telomere, whereas subTERRA-CUTs are transcribed toward the centromere [40,63]. (B) Schema of the X-element telomere; XC and XR regions are shown together with a general map of TERRA transcripts. The proposed approximative transcription start sites are at ~ 350 nts upstream of the telomeric repeat (T) and close to the end of the X-element [42]. Specific to the 730 bp long Tel15L X-element are the *RsaI* (R) and *HhaI* (Hh) restriction sites, the forwards ('a', 'b') and reverse ('c', 'd') oligonucleotide primers (**S2 Table**). Prior to this study, no sub-TERRA-CUTs were assigned to the Tel15L X-element.

membranes were probed with strand-specific oligonucleotides (**Fig 1A**; Y'F and Y'R), and the outcomes are shown in **Fig 2A** to **2C**. After treatment with T4 endo-V, changes in signal intensities of the bands (compare − and + lanes) reflect the number of CPDs that are present in the DNA fragments, soon after UV irradiation (0h) and during repair (0.5 to 4h). The formation and number of photoproducts depends on both, DNA sequence and DNA length. As expected, more CPDs formed in the ~4.7 kb than in the 1.3 kb fragment (compare **Fig 2A** and **2D** with **Fig 2C** and **2F**: 0h, '-' vs. '+' T4-V). In addition, the results show that NER proceeded at similar rate in both strands of the ~4.7 kb Y'-fragment. After 4 hours repair, ~90% of CPDs were removed from the C-rich strand and ~80% of CPDs were removed from the G-rich strand (**Fig 2A**). In the 3.0 kb fragment, comprised of about 88% of Y'-element sequences and 12% of telomeric repeats, the efficiency of NER was DNA strand dependent, as ~85% and ~60% of CPDs were removed from the C-rich and the G-rich strand, respectively (**Fig 2B**). Further information on NER in telomeric heterochromatin was obtained by applying the T4 endo-V assay to the 1.3 kb- terminal fragment, of which the telomeric repeats accounted for about 27% of the total length (**Fig 2C**). NER was three times more efficient in the C-rich than in the G-rich strand, since ~40% and ~13% of CPDs were removed from the former and the latter strand, respectively. Altogether, the data showed that there was DNA strand-specific NER in Y'-element telomeres. Control experiments performed with a yeast strain that lacked the essential NER protein Rad14 (*rad14Δ*) confirmed that CPDs were almost exclusively removed by NER (**S2 Fig**).

## TCR removes CPDs from both DNA strands of Y'-element telomeres

DNA strand-specific NER is well described for transcribed loci, where the transcribed strand is repaired faster than the non-transcribed strand by the Rad26-dependent TCR pathway [4,7]. Previously we focused on the uncharacterized, or dubious, ORFs that are present on both DNA strands of the Y' element [12]. To test the possibility that the centromere-to-telomere changes in NER efficiency was related to transcription of the uncharacterized ORFs, we followed repair in the *rad26Δ* strain, without separating the C- and G- rich DNA strand. The results were not conclusive [12], but they prompted us to analyze TCR in the C- and G- rich DNA strand, separately. Thus, we investigated if ncRNA transcription could foster TCR in Y' sub-telomeres and be responsible for distinct strand repair-efficiency at the ends of chromosomes (**Fig 2A** to **2C**). To differentiate between the participation of GGR and TCR, TCR deficient yeast (*rad26Δ*) were grown to early log phase, UV irradiated and incubated for various lengths of time, followed by the preparation of DNA as described above. The results showed that after 4 hours both strands of the ~4.7 kb fragment were repaired somewhat less efficiently in the absence of Rad26. About 63% of CPDs were removed from the two strands in the *rad26Δ* strain, whereas ~90% and ~80% of CPDs were removed from the C- and G- rich strand in WT cells, respectively (**Figs 2A, 2D, S3A** and **S3D**). For the 3.0 kb-fragment C-rich DNA strand, ~45% and ~85% of CPDs were repaired in *rad26Δ* and in WT cells, respectively (**Figs 2B, 2E** and **S3B**). In the G-rich strand, the difference in repair efficiency was slightly less pronounced: after 4 hours ~45% and ~60% of CPDs were removed in *rad26Δ* and WT cells, respectively (**Figs 2B, 2E**, and **S3E**). However, the participation of TCR was predominantly observed in the C-rich strand of the 1.3 kb-terminal fragment, where CPDs were twice less efficiently repaired in *rad26Δ* than in WT (**Figs 2C, 2F** and **S3C**). Remarkably, the G-rich strand of the same fragment was equally poorly repaired in both strains, as only about 15% of CPDs were removed after 4 hours. Hence, GGR and TCR removed CPDs from the two strands of Y'-telomeres, but TCR was more active than GGR in the C-rich strand and in the telomere distal regions. From these results we infer that transcription of the lncRNA TERRA promoted repair

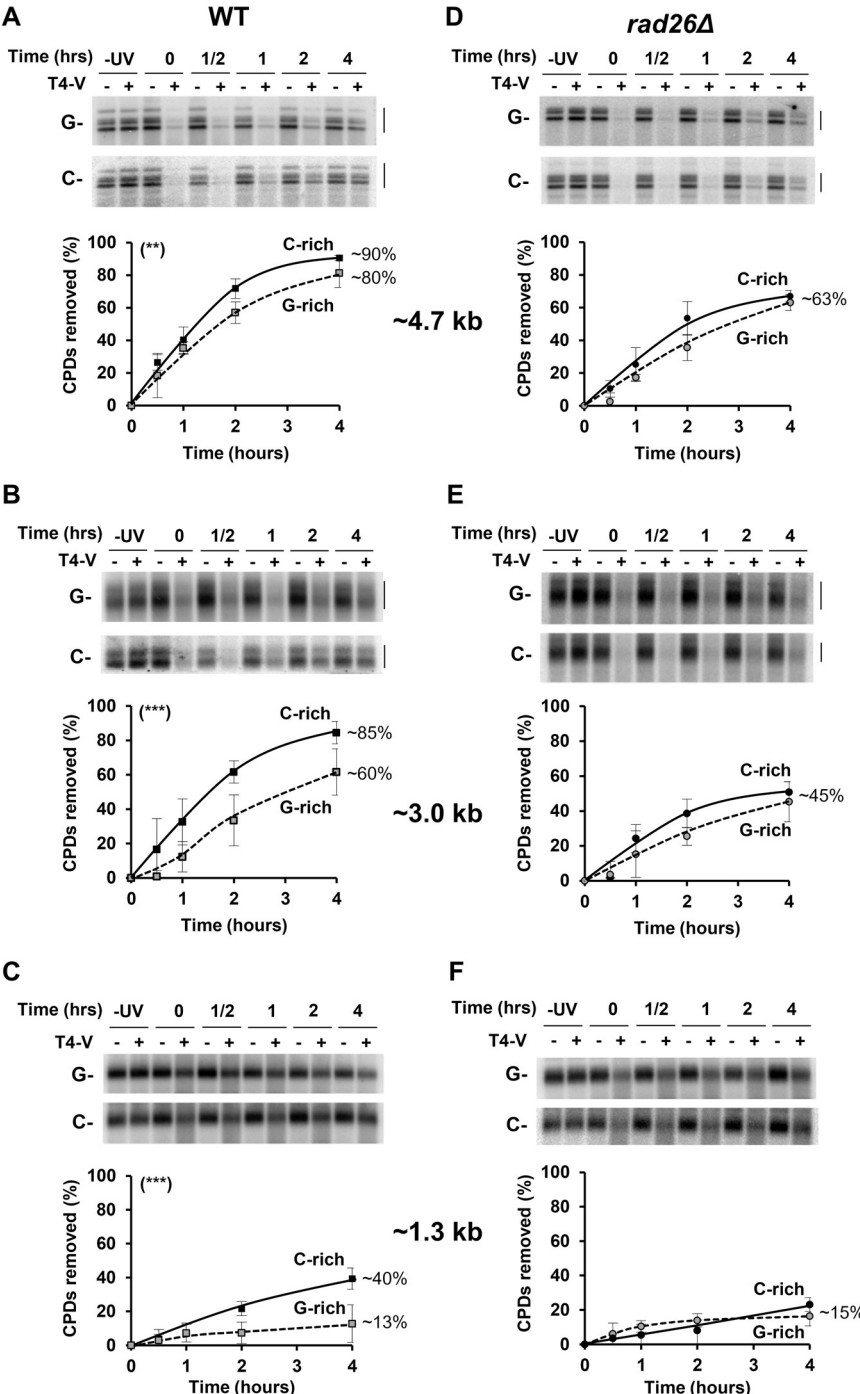

**Fig 2. NER in the Y'-element telomeres.** WT and *rad26Δ* yeast were irradiated at 180 J/m² and harvested at the indicated times in hours (hrs). Isolated DNA from non-irradiated (−UV) and irradiated cells (0 to 4 hrs) were digested with *HindIII*, *EcoRI* and *XhoI* (**Fig 1A**), and mock treated or treated with T4 endo-V (T4-V), denoted by − and +, respectively. After separation of the DNA fragments in 1% alkaline agarose-gels and blotting, the filter membranes were hybridized with single-stranded, strand-specific oligonucleotides (**Fig 1A** and **S2 Table**: Y'F and Y'R). The band signals correspond to the average of all Y'-elements in the cell. (A and D) Quantification of CPDs in the ~4.7 kb *HindIII/EcoRI* group of bands formed by the 4.4, 4.7 and 5.0 kb fragments (brackets). At 0 hour repair, on average for the C-rich strand 0.67 ± 0.07 CPD/kb, and for the G-rich strand 0.56 ± 0.09 CPD/kb. (B and E) Quantification of CPDs in the ~3.0 kb *HindIII* fragment, made of ~2.7 kb of Y'-element sequences and variable lengths (average 0.3 kb) of telomere repeats. At 0 hour repair, on average for the C-rich strand 0.69 ± 0.38 CPD/kb, and for the G-rich strand 0.43 ± 0.01 CPD/kb. Measurements were taken of the broad bands (brackets). (C and F) Quantification of CPDs in the

1.3 kb *XhoI* fragment, made of ~1 kb of Y'-element sequences and telomere repeats. At 0 hour repair, on average for the C-rich strand 0.58 ± 0.08 CPD/kb, and for the G-rich strand 0.36 ± 0.1 CPD/kb. Data are from quantification of phosphor images for the C-rich (C-) and G-rich (G-) strands of WT (A-C) and *rad26Δ* (D-F). The means ±1SD are of 3 independent experiments. *P*-values were calculated using a 2way ANOVA test: (**) p < 0.01, (***) p <0.001.

of CPDs in the C-rich strand of the terminal 1.3 kb fragment, where GGR was ineffective (**Figs 2C, 2F** and **S3C**). In fact, the participation of TCR paralleled the transcription map and density of all lncRNAs (**Fig 1A**). There is evidence suggesting that most of the C-rich strand could be transcribed into 0.1–1.2 kb long TERRA and subTERRA-Xuts, while the G-rich strand is only weakly transcribed into subTERRA-Cuts [40]. Lastly, the data show that the efficiency of NER gradually decreased in a centromere to telomere direction (compare **Fig 2A** and **2D** with **Fig 2B and 2E**, and **Fig 2C and 2F**). This outcome reflected the percent of telomeric repeats, and therefore of silenced chromatin, in the ~4.7, 3.0 and 1.3 kb DNA fragments, which is consistent with our previous findings demonstrating that silenced chromatin hampered NER [12]. The efficiency of NER has been previously compared between DNA fragments of different lengths [25], as the number of PDs and NER complexes is proportional to length of the DNA fragments.

## In the Tel15L X-element, TCR occurs only in restricted segments of the C-rich DNA strand

To determine if ncRNA transcription promotes repair of pyrimidine dimers (CPDs and 6,4-PPs) in silenced heterochromatin of the X-element, WT and *rad26Δ* strains were UV irradiated and incubated for different times to allow DNA repair. Isolated DNA was digested with the indicated restriction enzymes (**Fig 1B**) and the presence of PDs was analyzed at nucleotide resolution by a primer extension assay that is based on efficient and precise blockage of the Taq polymerase by PDs ([41], see Materials and Methods). The DNA-primers were designed to investigate NER in the X-element of chromosome 15-L (**Fig 1B** and **S2 Table**), and representative sequencing gels of primer-extension products for the G-rich and C-rich strand are shown in **Figs 3** and **4**, respectively. The top bands correspond to full length-extension of undamaged DNA fragments, the bands below indicate the presence of PDs and their positions are determined by DNA sequencing lanes (not presented). The band intensity is relative to the frequency at which PDs form [compare -UV (U) with +UV (0h repair)] and decrease of the intensity is proportional to the efficiency at which the corresponding PD is repaired (compare 0h with 4h repair). After correction for variations occurred during gel loading of DNA samples, by normalizing the intensity of each band to the sum of all bands in a lane, quantifications were done as described in 'Materials and Methods'. The data obtained with *rad26Δ* cells were plotted as percent of repair over time and compared to the data obtained with WT cells, for single PDs (**S4**–**S7 Figs**) and for clusters of PDs (**Figs 3C** and **4C**). The results show that the repair rates of PDs in the G-rich strand were very similar for the two strains (**Figs 3C** and **S4** and **S5**) and that, consistently with the very low frequency of potential PD formation-sites at the 3'-end of the XR-region only one UV photoproduct (PD 27; cluster ix) formed in this portion of the G-rich strand (**Figs 3** and **S5**). Similarly, repair of most PDs in the C-rich strand was not affected when Rad26 was absent, except for PDs -1 to +8 (clusters -iv, -v and -vi) that were repaired less efficiently in *rad26Δ* (**Figs 4C** and **S6** and **S7**). The PDs 1 to 5 were slowly removed by TCR, possibly because transcription was partially hindered in silenced heterochromatin (see below). Hence the results indicated that in WT cells, TCR operated in a short fragment of about 105 bases at the beginning of the Tel15L X-element, C-rich DNA strand.

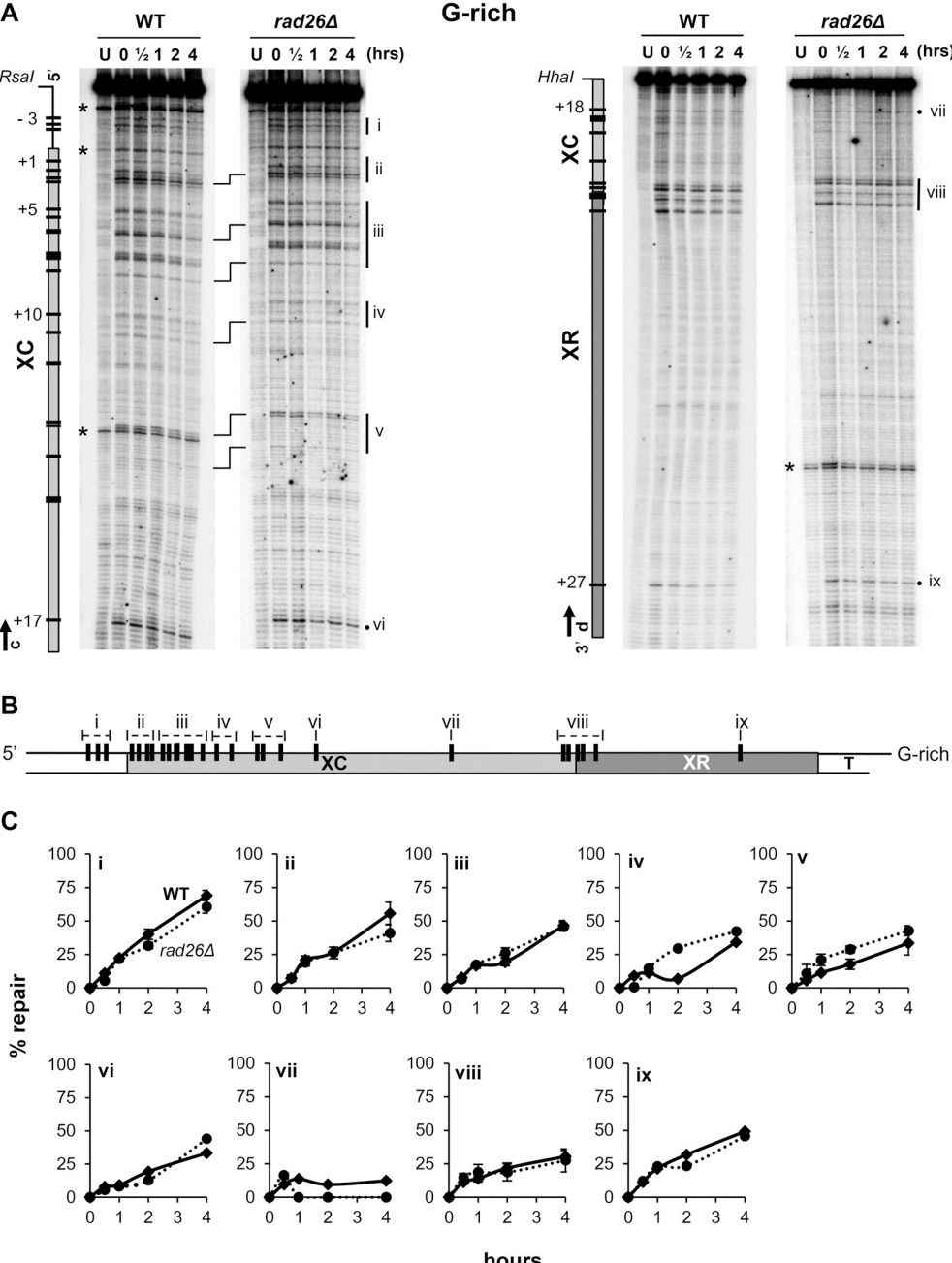

**Fig 3. Repair of pyrimidine dimers in the Tel15L X-element of WT and *rad26Δ* cells, G-rich strand.** Yeast were UV irradiated at 180 J/m² and harvested after 0, 0.5, 1, 2 and 4 hours (hrs). Isolated DNA from non-irradiated (U) and irradiated cells (0 to 4 hrs) were digested with *RsaI* or *HhaI* and used as template for TaqI polymerase primer-extension. (A) Repair of PDs was assessed by the extension of primers 'c' and 'd' (arrows) (S2 Table). Maps on the left of sequencing gels are of the X-element with the 5'- and 3'- ends, its XC- (light gray) and XR- (dark gray) regions, and flanking DNA (line). PDs that were quantified and plotted in S4 and S5 Figs are represented by black bars and numbered negatively or positively when present outside or inside of the X-element, respectively. The position of PDs was verified by DNA sequencing lanes (not included in the figure). Asterisks point to natural sequences causing TaqI polymerase arrest, brackets align bands between gels and vertical bars mark PD clusters ('i' to 'ix'). (B) General map of the X-element (~730 bp) with the position of PDs (black bars) that approximatively reproduce the migration of bands in the gels. (C) Repair plots (percent of repair over time) are for clusters of PDs ('i' to 'ix'); WT (diamond, continuous line) and *rad26Δ* (circle, dotted line). The means ±1SE are for clusters of at least 3 PDs. *P*-values were calculated using a 2way ANOVA test using a Šídák's multiple comparisons test. *P*-values show no significant statistical difference.

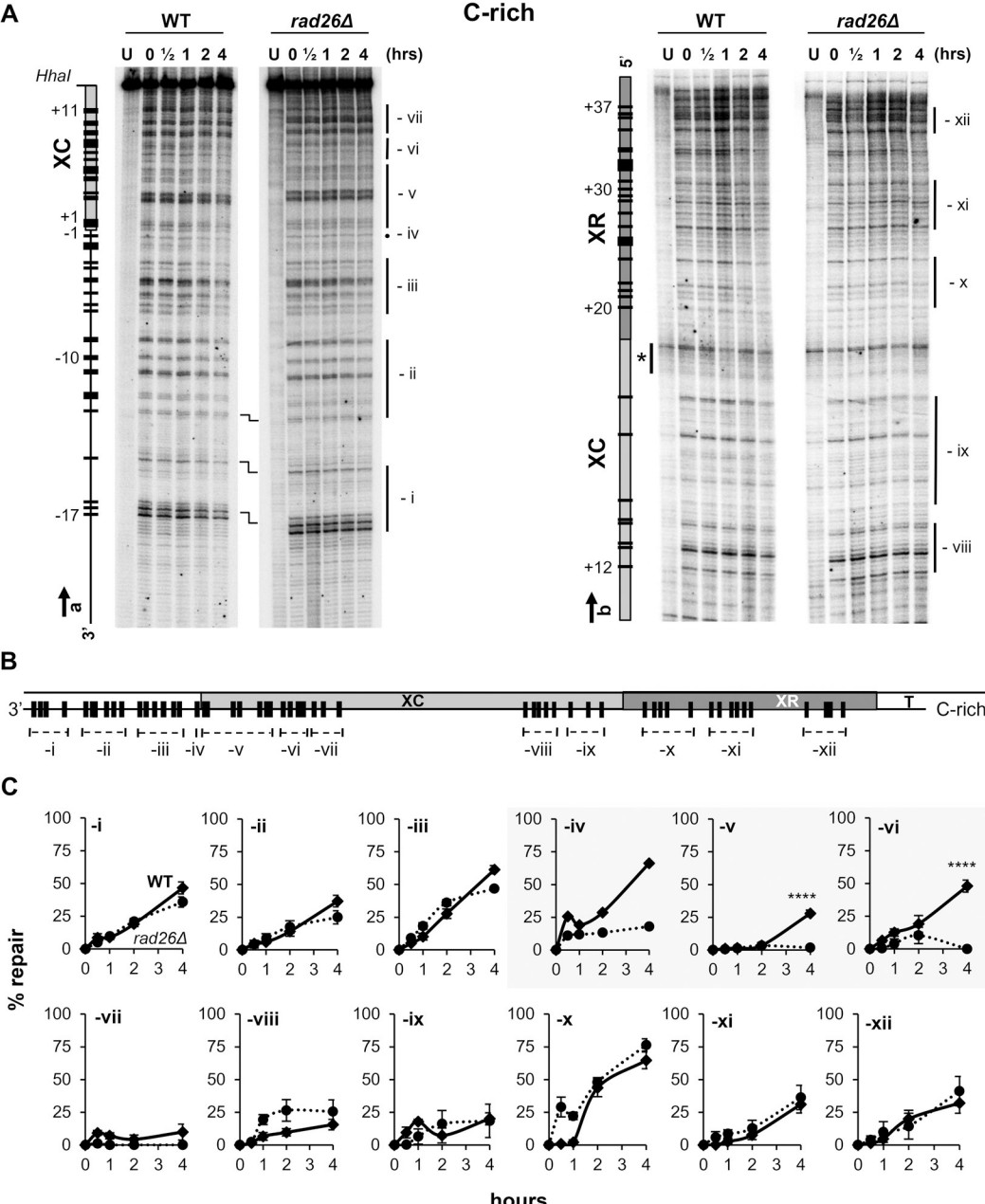

**Fig 4. Repair of pyrimidine dimers in the Tel15L X-element of WT and *rad26Δ* cells, C-rich strand.** Yeast were UV irradiated as described in **Fig 3**. Isolated DNA from non-irradiated (U) and irradiated cells (0 to 4 hrs) were digested with *Hha*I and used as template for TaqI polymerase primer-extension. (A) Repair of PDs was assessed by the extension of primers 'a' and 'b' (arrows) (**S2 Table**). Description of maps on the left of sequencing gels is as in **Fig 3**. PDs that were quantified and plotted in **S6** and **S7 Figs** are represented by black bars and numbered negatively or positively when present outside or inside of the X-element, respectively. The position of PDs was verified by DNA sequencing lanes (not included in the figure). Symbols are as described in **Fig 3**, and vertical bars mark PD clusters ('-i' to '-xii'). (B) General map of the X-element (~730 bp) with the position of PDs (black bars) that approximatively reproduce the migration of bands in the gels. (C) Repair plots (percent of repair over time) are for clusters of PDs ('-i' to '-xii'); WT (diamond, continuous line) and *rad26Δ* (circle, dotted line). The means ±1SE are for clusters of at least 3 PDs. *P*-values were calculated using a 2way ANOVA test using a Šídák's multiple comparisons test. *P*-values are indicated for the 4 h repair time point: (****) p < 0.0001. ANOVA test cannot be applied to cluster–iv that comprises only 1 PD.

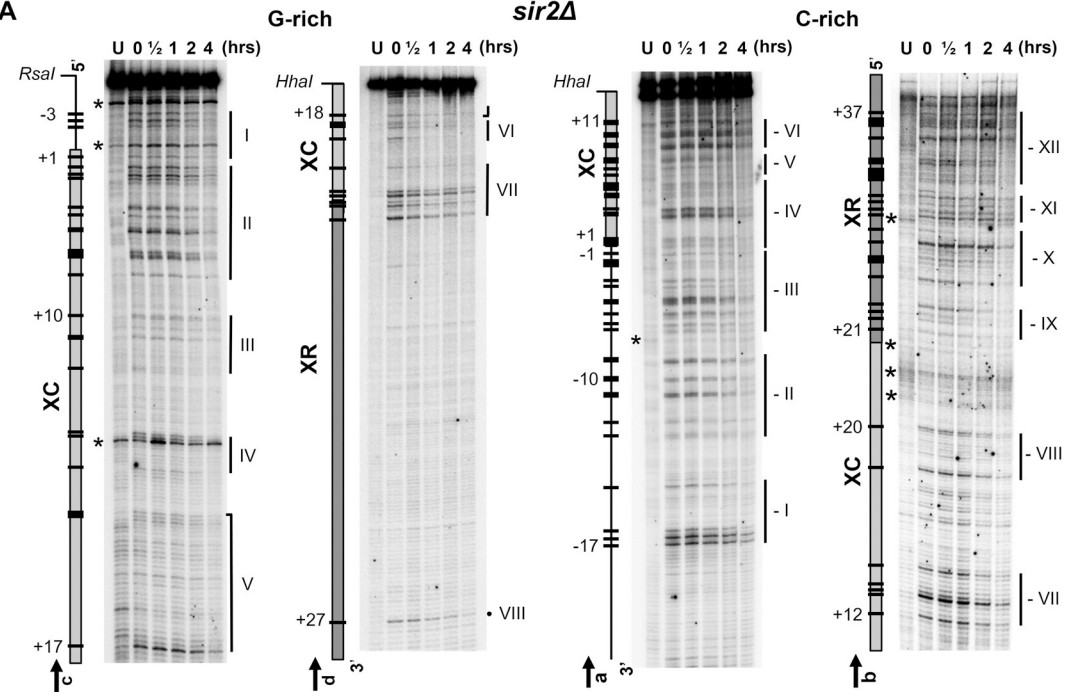

**Fig 5. Mapping of pyrimidine dimers in the Tel15L X-element of *sir2Δ* cells at nucleotide resolution.** The DNA was prepared and assessed for the presence of PDs as described in **Figs 3** and **4**. Description of maps on the left of sequencing gels is as in the previous figure legends. PDs that were quantified and plotted in **S8**, **S9**, **S10** and **S11 Figs** are represented by black bars and numbered negatively or positively when present outside or inside of the X-element, respectively. The position of PDs was verified by DNA sequencing lanes (not included in the figure). Symbols are as described in the previous figure legends, and vertical bars mark PD clusters ('I' to 'VIII' and '-I' to '-XII').

## Sir2 delays NER in Tel15L X-element chromatin

Sir proteins are present on X-elements and telomeric repeats [23,24]. We previously showed that PDs were slowly removed from the Tel15L X-element of WT cells, and that Sir3 of the Sir2, 3, 4 complex affected the efficiency of NER [12]. In the present study we investigate if the Sir complex altered the efficiency of one or both of the NER sub-pathways, namely GGR and or TCR. Initially, the repair of PDs was followed in *sir2Δ* cells at nucleotide resolution (**Fig 5**) and the resulting data were plotted against the data obtained with WT cells (**Fig 6**). The outcomes showed that numerous PDs, but not all, were repaired faster in the absence of Sir2, albeit to different degrees depending on their position in the X-element (**S8–S11 Figs**). For both strains, the clusters were redefined from the PDs described in **S8–S11 Figs** (G-rich strand: I to VIII; C-rich strand: -I to -XII). As illustrated in **Fig 6B** and **6C**, the efficiency of NER was clearly enhanced in chromatin lacking Sir2, except distant upstream of the X-element (cluster -I), at the beginning of the XC-region (opposite clusters: -III, I and -V, III) and at the end of the same region (cluster VI), where nearly comparable efficient repair rates were measured in WT and *sir2Δ* strains. These results suggested that silenced heterochromatin was not homogeneously spread over the X-element, leaving gaps that were favorable to transcription and thus TCR. As shown in **Fig 7A**, TCR removed PDs in clusters -III and -V two times faster than PDs in cluster -IV (**Fig 4C**, compare the repair of clusters -iv and -vi with cluster -v), where there was a predominant participation of GGR. In general, GGR was the main sub-pathway that restored the Tel15L X-element DNA after UV irradiation, and it was considerably hindered by the presence of Sir2 (**Fig 7A**).

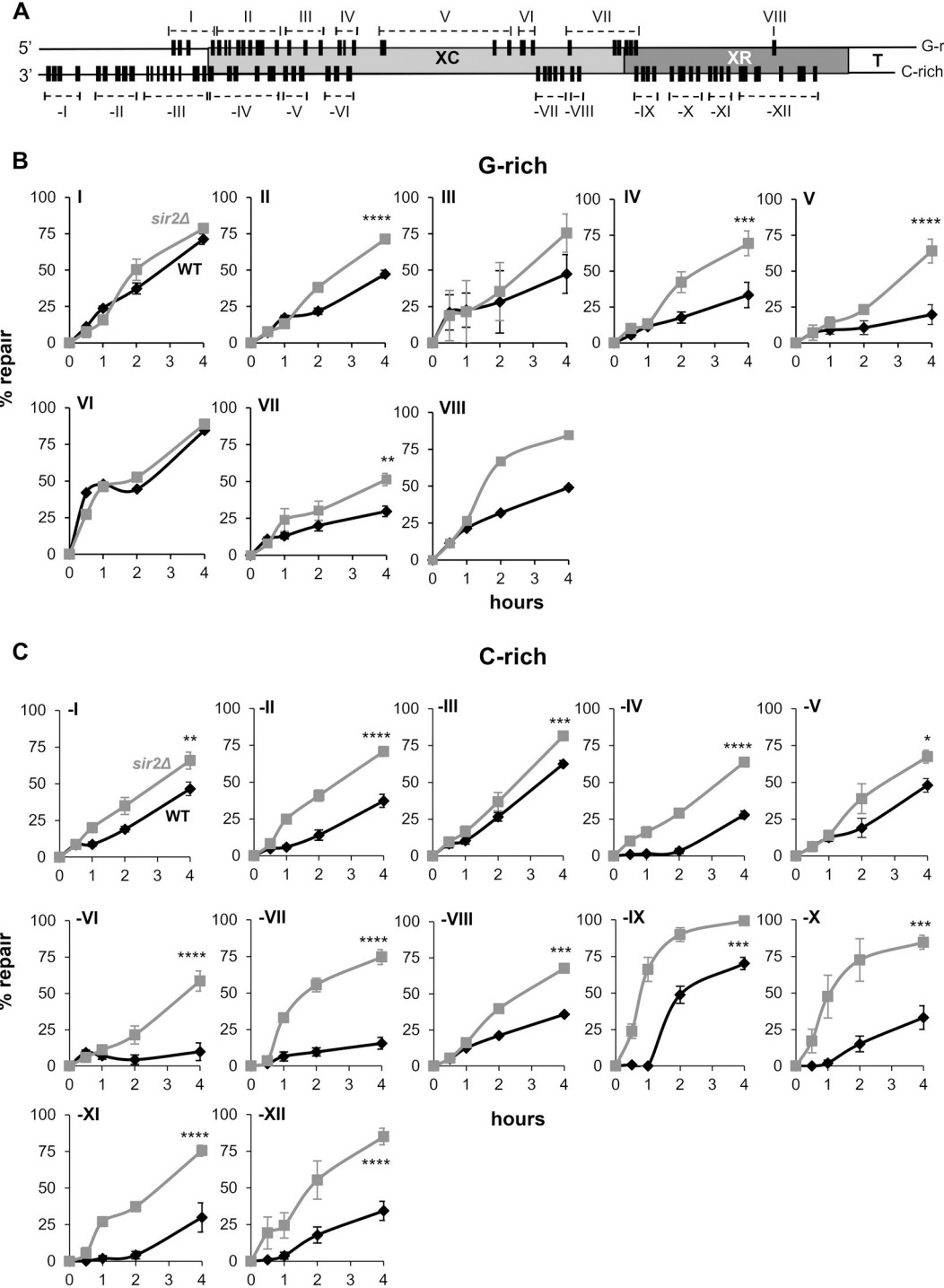

**Fig 6. Repair of pyrimidine dimers in the Tel15L X-element of *sir2Δ* cells.** (A) General map of the X-element (~730 bp) with the position of PDs (black bars), in the G- and C-rich strand, that approximatively reproduce the migration of bands in the gels of **Fig 5**. (B) and (C) Repair plots (percent of repair over time) are for clusters of PDs in the G- and C-rich strand, respectively. WT (diamond, black line) and *sir2Δ* (square, grey line); the means ±1SE are for clusters of at least 3 PDs. *P*-values were calculated using a 2way ANOVA test using a Šídák's multiple comparisons test. *P*-values are indicated for the 4 h repair time point: (*) p < 0.05, (**) p < 0.01, (***) p <0.001, (****) p < 0.0001. ANOVA test cannot be applied to cluster VIII that comprises only 1 PD.

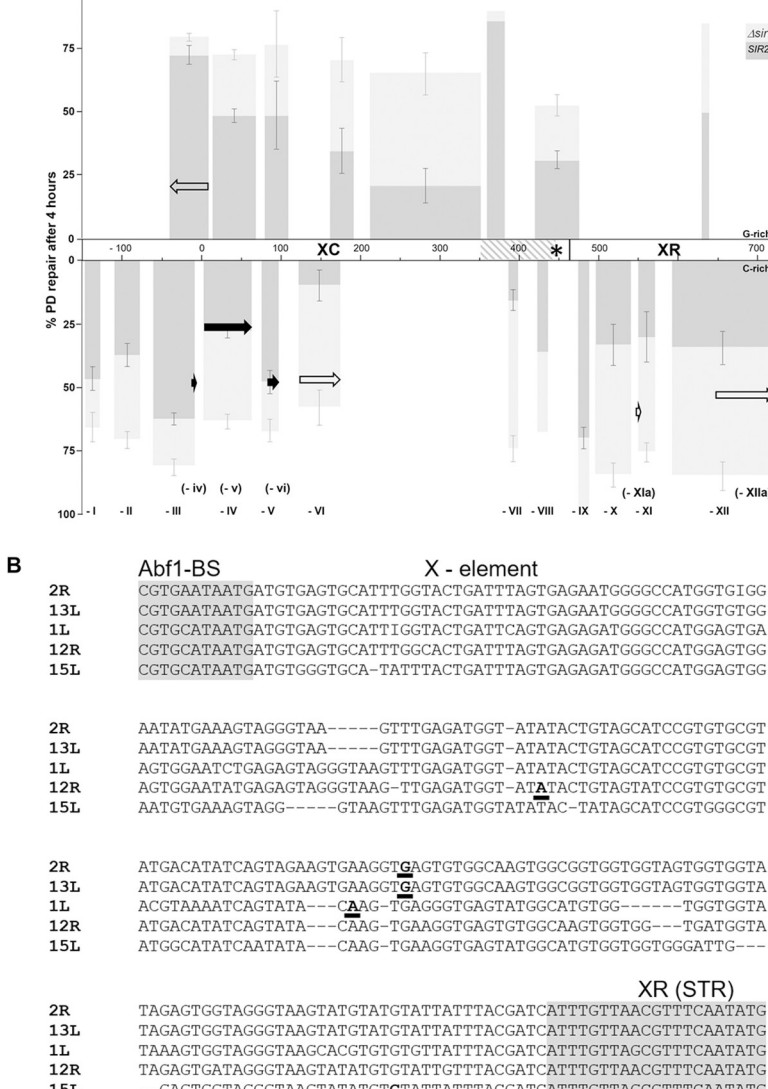

**Fig 7. TCR assists GGR to remove PDs in Tel15L X-element silenced heterochromatin.** (A) Column chart summarizing repair of PD clusters (roman numerals) in the G- and C-rich strand. The percent of repair after 4 hours was extracted from the repair plots for WT, *rad26Δ*, *sir2Δ* and *sir2Δrad26Δ* cells (**Figs 3**, **4**, **6** and **9**) and plotted against the Tel15L X-element DNA sequence. Dark and light grey columns denote the average repair of PDs by NER (GGR + TCR), in Sir2 containing (*SIR2⁺*) and depleted (*sir2Δ*) chromatin, respectively. Error bars (± 1SE) were calculated for clusters of 3 or more PDs. Black and white arrows denote the average repair of PDs by TCR, in *SIR2⁺* and *sir2Δ* cells respectively. The shadowed area at the end of the XC region represents the domain where the transcription start-sites (TTSs) were mapped for 5 telomeres, see (B). The star points to the TERRA TSS for the Telomere 15L. In the C-rich strand, between clusters -VI and -VII the assay could not be applied to identify PDs because primers cross-hybridized with unspecific repetitive DNA sequences or could not be extended by the TaqI polymerase. In the G-rich strand of the XR-region, only cluster VIII (PD 27; +635) was identified by the primer extension assay, reflecting the low content of potential PD forming sites in the DNA sequence. (B) DNA sequence similarities between the X-elements of Telomere 2R, 13L, 1L, 12R and 15L. The Abf1 binding site and the XR (STR) regions are in grey. Nucleotides in bold correspond to the TERRA transcription start-sites (TSS).

Previously, the TERRA transcription-start site (TSS) was mapped within 346 nucleotides upstream of the Tel1L X-element TG-repeat, close to the proximal-end of the XC-region [**42**]. In addition, we mapped the TERRA TSSs for the X-elements of Tel 2R, 15L, 12R and 13L of

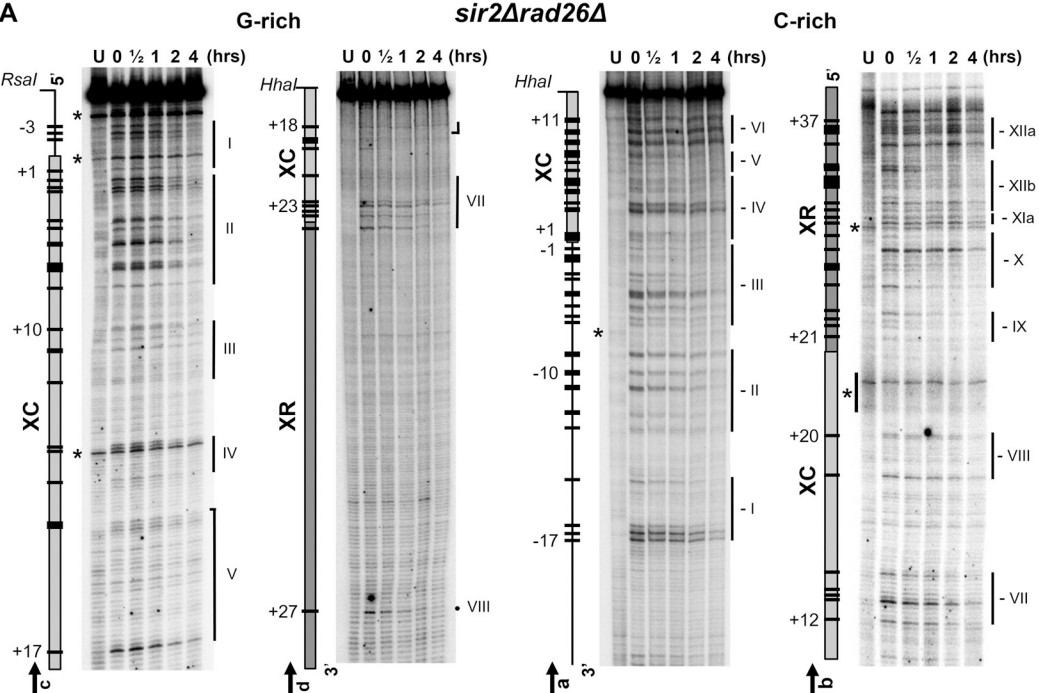

**Fig 8. Mapping of pyrimidine dimers in the Tel15L X-element of *sir2Δrad26Δ* cells at nucleotide resolution.** The DNA was prepared and assessed for the presence of PDs as described in **Figs 3** and **4**. Description of maps on the left of sequencing gels is as in the previous figure legends. PDs that were quantified and plotted in **S13–S16 Figs** are represented by black bars and numbered negatively or positively when present outside or inside of the X-element, respectively. The position of PDs was verified by DNA sequencing lanes (not included in the figure). Symbols are as described in the previous figure legends, and vertical bars mark PD clusters ('I' to 'VIII' and '-I' to '-XII').

the *sir2Δ* strain (**S12 Fig**). The sequences of the five X-element proximal ends are highly similar (**Fig 7B**), and all these TERRA TSS were mapped within about 100 bases from the end of the XC region (**Fig 7A** and **7B**). Interestingly, PDs 19 and 20 (cluster VI) that fell in this region were effectively removed in WT cells, and the efficiency of GGR did not improve in the absence of Sir2 (**Figs 6B** and **7A**). These results suggested that the chromatin surrounding cluster VI was loosened, allowing fast repair and being potentially favorable to transcription initiation. In WT (*SIR2*[+]) cells, TCR was not observed close to the telomere proximal-end of the XC-region, nor downstream in the XR-region (**Figs 4** and **7A**), likely because the presumed transcription was too low to support TCR. This is supported by previous studies, showing that in the presence of Sir proteins, very little TERRA transcription was measured in the Tel 1L XR-region [42,43]. Summing up, in *SIR2*[+] yeast only PDs of clusters -iv to -vi were primarily removed by TCR, whereas PDs that formed outside of these clusters were merely repaired by GGR. The presence of Sir2 significantly impeded NER, except in three chromatin domains surrounding the PD-clusters -III/I, -V/III and VI, where there could be gaps in the silenced chromatin.

Consistent with their function in establishing silenced chromatin, Sir proteins repress TERRA transcription [42,43]. Consequently, efficient repair of PDs in the Tel15L X-element of *sir2Δ* yeast (**Figs 6** and **S8–S11**) could have resulted from efficient GGR, due to a more accessible heterochromatin, or to TERRA transcription-driven TCR. To evaluate these possibilities, NER was assessed in the *sir2Δrad26Δ* double mutant, and representative sequencing gels of primer-extension products are shown in **Fig 8**. The resulting measurements of repair for each PD (**S13–S16 Figs**), and for clusters of PDs (**Fig 9**), were compared to the

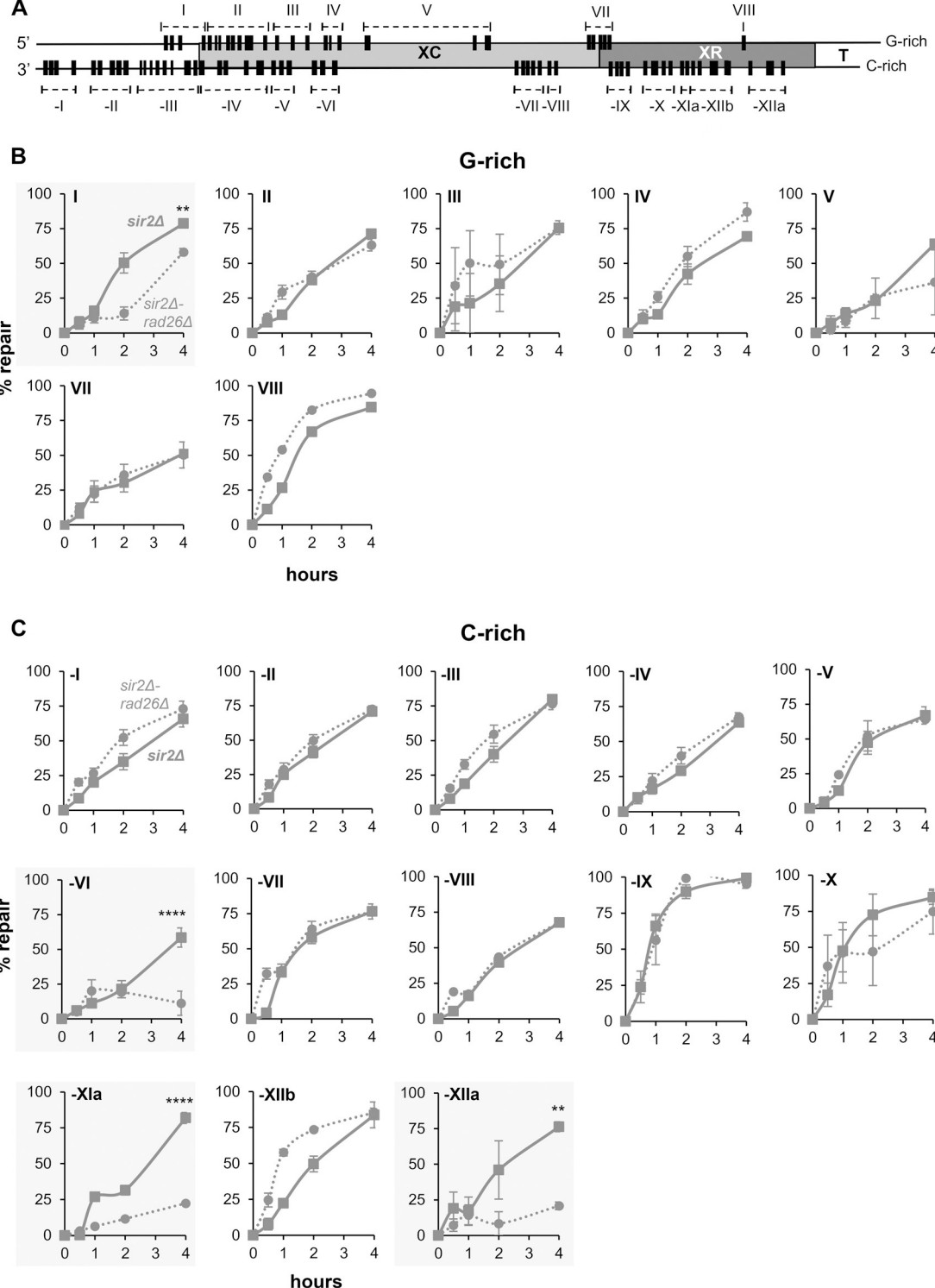

**Fig 9. Repair of pyrimidine dimers in the Tel15L X-element of *sir2Δrad26Δ* cells.** (A) General map of the X-element (~730 bp) with the position of PDs (black bars), in the G- and C-rich strand, that approximatively reproduce the migration of bands in the gels of **Fig 8**. (B) and (C) Repair plots (percent of repair over time) are for clusters of PDs in the G- and C-rich strand, respectively. *sir2Δ* (square, continuous line) and *sir2Δrad26Δ* (circle, dotted line); the means ± 1SE are for clusters of at least 3 PDs. *P*-values were calculated using a 2way ANOVA test using a Šídák's multiple comparisons test. *P*-values are indicated for the 4 h repair time point: (**) p < 0.01, (****) p < 0.0001. ANOVA test cannot be applied to cluster VIII that comprises only 1 PD.

measurements obtained with the *sir2Δ* strain. In *sir2Δ* yeast, efficient repair of most PDs on both DNA strands was not affected by the absence of Rad26. Some TCR was identified over about 50 nts in the G-rich strand, but GGR remained the principal pathway that removed PDs -3 to +1 (cluster I) (**Figs 7A** and **S13**). These results implied that at the beginning of the X-element and in the absence of Sir2, some lncRNA transcription occurred in the opposite direction of TERRA. In the C-rich strand, TCR removed DNA lesions over about 50 nts, including PDs 9 to 11 (cluster -VI) (**Figs 7A** and **9C** and **S16**). On the other hand, flanking PDs -1 to +8 (clusters -iv, -v and -vi) that in WT (*SIR2+*) were largely removed by TCR (**S6 Fig**), in *sir2Δ* cells they were removed by GGR (**S15 Fig**). These outcomes suggested that the extent and location of lncRNA transcription depended on the presence of Sir2.

In the absence of Sir2, the increase in TERRA transcription allowed to map the TSS at the 3'-end of the XC-region of Tel 15L (**Fig 7A**, star; **Figs 7B** and **S12**). Interestingly, TCR was measured downstream of the TSS for PDs 29 and 30 (cluster -XIa) and PD 35 to 37 (cluster -XIIa) (**Figs 7A**, **9C** and **S16**). We suggest that the discontinuing participation of TCR (between clusters -XIa and -XIIa) could result from the local binding of proteins that, with the Sir complex, form the telomeres. In the absence of Sir2, TERRA transcription promotes TCR, but GGR is still required to remove DNA damage in the presence of telomere proteins. The primer extension assay could not be applied to measure repair from nt 174 to nt 387 (**Fig 7A**), because the repetitive sequence of the C-rich strand hampered the Taq polymerase to faithfully extend oligo-nucleotide primers. Thus, it remains unknown if TCR stretched further downstream into the XC-region.

## Discussion

The long-lasting tenet that telomeric regions were transcriptionally silent predicted that GGR removed PDs from the chromosome ends [12]. Although these heterochromatic regions were supposedly not accessible to the transcription machinery, more recent studies showed that RNA polymerase-II was present on the sequence and that transcription took place, albeit at extremely low levels [44]. For yeast, there is increasing evidence that all telomeres can generate lncRNAs called TERRA. Additionally, Y'-type telomeres are transcribed into Sub-TERRA-X-UTs and, in the opposite direction of TERRA, into Sub-TERRA-CUTs [36,40,45]. Analyses based on deep-sequencing of RNAs derived from Y'-elements suggest that Sub-TERRA-XUTs are much more abundant than sub-TERRA-CUTs [40]. All these findings prompted us to investigate if TCR, the NER sub-pathway that specifically removes PDs from the transcribed strand of active genes, could participate with GGR in the repair of UV induced DNA damage in Y'-telomeres. Our experiments show that the efficiency of NER decreases in a centromere-to-telomere direction. In the most distal ~4.7 kb fragment, CPDs are effectively repaired, like CPDs in the transcriptionally active mating-type locus (*MATa*) or in the multiple copies of rRNA genes [12,25], and far more proficiently than CPDs in silenced *HML* and *HMR* loci [12]. Moreover, similar efficient repair occurs in the terminal ~3.0 kb fragment, reflecting its high percent (~90%) of the complex Y' DNA-sequences. The CPDs are equally repaired in both strands of the ~4.7 kb fragment, whereas in the 3.0 kb fragment the C-rich strand is repaired faster than the G-rich strand. Consistently, transcription of lncRNA is more significant in the C-rich than in the G-rich strand [40] and, in TCR-deficient cells, the repair of CPDs is primarily affected in the C-rich strand. Nonetheless TCR occurs in both DNA strands, contributing to the rather high repair rates in Y'-elements. Thus, the lack of Sir proteins [23,24], the occurrence of RNA polymerase-II transcription in both DNA strands [40] and the efficient repair of UV induced DNA lesions by GGR and TCR, all would indicate that Y'-elements have a relatively open chromatin structure. On the contrary, the structure of the most

distal telomeric DNA is described as a dense nucleoprotein structure that include Rap1, Sir-2, -3 and -4, as well as Rif-1 and -2, to protect the chromosome ends from being recognized as double-strand breaks, from unwanted recombination and from chromosome fusions [20]. In agreement with the protective function of telomeres, the results obtained with the *rad26Δ* strain show that GGR is inefficient in the terminal 1.3 kb fragment, which is made of about one third of telomeric repeats. Whatever structure or mechanism that impedes GGR, resulting in the repair of only ~15% of CPDs after 4 hours, it is in part overcome by TCR that allows removal of up to ~40% of CPDs from the C-rich strand of WT cells. This set of experiments indicate that lncRNA-transcription fosters repair of UV induced DNA lesions, complementing efficient GGR and assisting inefficient GGR in Y'-element chromatin.

Nucleosomes and Sir proteins form a heterochromatin-like structure on the X-elements [23,24]. GGR slowly removes PDs from the Tel15L X-element and, on average, after 4 hours less than about 46% and 38% of PDs are removed from the G-rich and C-rich strand, respectively. The implication of silenced heterochromatin in the slow repair of PDs is confirmed by the increased efficiency of NER in *sir2Δ* yeast, where the average repair improves to about 69% and 74% for the respective DNA strands. These results agree with our previous study, where we suggested that the stabilization of nucleosomes by Sir3 might inhibit NER [12]. This interpretation of the data derived from the indication that PDs could be exposed to NER during transient shifts in nucleosome positioning [9]. Thus, slow repair in the X-element heterochromatin could result from the reduced mobility of nucleosomes. Although, GGR is the dedicated sub-pathway in the X-element heterochromatin [12], TCR removes PDs from about 120 nucleotides of the C-rich strand, assisting the otherwise almost completely inefficient GGR (**Fig 7A**). Hence TCR occurs at the beginning of the XC-region, but we cannot exclude that it happens further downstream, because the primer-extension assay could not be applied between nts +174 and +387. But TERRA transcription is extremely low [42,43] and it may not always be sufficient to support TCR, particularly in silenced chromatin domains. Then, TCR is observed into the XC-region and in the XR-region only after releasing the Sir2 inhibition of TERRA transcription (**Fig 7A**). In the WT (*SIR2*+), repair analyses at the nucleotide level point out that TCR is primarily operational in two short segments, where the efficiency of repair is only marginally improved when Sir2 is absent (*sir2Δ*). Thus, we suggest that there could be gaps in silenced chromatin that permit transcription and, therefore, TCR. In the absence of Sir2, the repair of PDs does not improve in a third short segment that is located within a region comprising the TSSs of TERRA transcripts (**Fig 7A**, shaded area). This correlation strengthens the hypothesis that there could be a link between chromatin gaps and lncRNA transcription. Hence, given the heterochromatic structure of telomeres, perhaps near the TERRA transcription start sites the chromatin is open and poised for transcription [46]. While at present, there is no evidence for lncRNA transcription of X-elements in a telomere-to-centromere direction, our results further suggest that there could be Sub-TERRA-CUTS at the beginning of the X-elements, at least in the absence of Sir2. Finally, it is worth noting that these experiments examined only one specific X-telomere. Consequently, if very low transcription is detected at a pan-telomere level or at the Y'-telomeres (about half of all telomeres), the specific transcription of a single X-telomere must be extremely low. Because our experiments still detect TCR in some segments of Tel15L X-element is, therefore, a very significant indication of its presence.

In human cells, the estimate of NER efficiency at telomeres is controversial. An early study indicated that cells efficiently removed PDs from telomeric DNA [47]. This conclusion was supported by findings obtained with fibroblasts expressing exogenous telomerase, which showed that CPDs in telomeres were repaired faster than CPDs in the bulk of the genome [48]. In contrast, Rochette et al. [49] found that CPDs are not repaired in telomeres and that cells tolerate persistent high levels of damaged telomeric DNA. Albeit at slow rate, our results

indicate that PDs in yeast telomeres are mostly repaired by GGR, and that Rad26 driven TCR assist GGR in specific short telomeric regions. *RAD26* is the yeast homolog of the human Cockayne's syndrome (CSB) gene [6]. Mutations in *CSB* cause a degenerative disorder, which is characterized by the lack or repair in the transcribed strand of active genes [2–5]. Pervasive transcription of non-coding regions throughout the genome could underscore the importance of TCR and its relevance for the related disease in human.

The ends of chromosomes are transcribed into different species of ncRNA, including TERRA. Their functions are not completely understood, at least in part because they are transcribed at a very low level and then rapidly degraded. Moreover, it is difficult to demonstrate the expression of specific ncRNAs at unique telomeres, since oligonucleotide probes designed for this purpose often show cross-hybridization or no signal [40]. It is proposed that ncRNAs participate in the regulation of telomere maintenance by telomerase and homologous recombination, in the changes in chromatin composition and telomere mobility, and in the telomeric DNA damage-response [50,51]. The recent studies by d'Adda di Fagagna and colleagues reported that induction of double strand breaks (DSBs) led to the assembly of functional transcriptional promoters at DNA damage sites. At telomeres of mammalian cells, damage-induced lncRNAs resulted from transcription of both C- and G-rich strands [52]. After processing to short RNAs, damage-induced lncRNAs recruited DNA damage response proteins into foci [53]. It will be interesting to search if these mechanisms are unique to DSBs, or if other type of DNA damage could induce lncRNA transcripts.

The heterochromatin-like structure of sub-telomeres predicted that PDs were mostly removed by GGR, in agreement with our findings. TCR highly correlates with the rate of transcription [54], and TERRA transcription is extremely low in both the X-element, where it is repressed by Sir2/3/4, and in the Y'-element where it is repressed by the Rap1 recruited Rif1/2 complex [43]. Consequently, in the two sub-telomeric regions there is only a modest participation of Rad26 dependent NER. Nonetheless, here we describe that ncRNA transcription at chromosome ends foster TCR in heterochromatic regions, complementing the activity of GGR and assisting inefficient GGR that otherwise would leave unrepaired UV photoproducts. Because the modulation of NER efficiency can provide indirect information on the structure of chromatin [55,56], our results also suggest that silenced heterochromatin is not homogeneously spread over the Tel15L X-element and that there could be open domains (gaps) where ncRNA transcription can initiate and take place.

## Materials and methods

### Yeast strains, media and growth conditions

Information on the BY4741 strains, derived deletion mutants and DNA sequence for TEL15L (SGD ID:S000028929) and for TEL01L (SGD ID:S000028862) are available on the *Saccharomyces Genome Deletion Project* web site [http://www-sequence.stanford.edu/group/yeast_deletion_project/deletions3.html] [57], and *Saccharomyces genome database* web site (SGD) https://www.yeastgenome.org (**S1 Table**). Yeast cultures were grown to log-phase (~$10^7$ cells/ml) in yeast extract-peptone-dextrose (YEPD) at 30°C, under continuous rotation. After centrifugation, cells were re-suspended in ice-cold PBS (137 mM NaCl, 2.7 mM KCl, 1.76 mM KH$_2$PO$_4$,10mMNa$_2$HPO$_4$, pH 7.0) to the final concentration of $2 \times 10^7$ cells/ml. Cell suspensions were poured into trays to a depth of ~1 mm and irradiated with a UV dose of 180 J/m$^2$ (primary 254 nm), measured with a UVX radiometer (Ultra-Violet Products, Upland, USA). Yeast cells were harvested, re-suspended in pre-warmed YEPD and incubated in the dark at 30°C with continuous shaking for different repair times, as indicated in the text.

## DNA extraction

For each repair time point, ~2 x $10^9$ cells were collected, and the DNA was extracted from isolated nuclei, as described in [12]. The DNA samples were re-suspended in TE buffer and quantified with a fluorimeter (Dyna Quant 200, Amersham), before digestion with the appropriate restriction enzymes, as indicated in the text.

## T4 endonuclease V, Alkaline Gel Electrophoresis and Southern Blotting

DNA samples were incubated with the enzyme T4 endonuclease V (T4 endo-V; Epicentre) according to the manufacturer's recommendations. After T4 endo-V treatment, ~ 1 μg of each DNA sample were separated on denaturing 1% alkaline agarose-gels and transferred to Hybond XL membranes (GE-Healthcare) in 0.4 N NaOH. The probes were generated by $^{32}$P-endlabeling specific oligo nucleotides (**S2 Table**). Pre-hybridization, hybridization and washing were done at 65˚C as previously described [58].

## Primer extension

Labeling of single stranded DNA oligonucleotides and primer extension assays were performed essentially as described in [12]. After denaturation (95˚C, for 10 min), DNA samples were chilled 10 min on ice before adding 0.25U of Taq polymerase. The primer extension reaction was done as following: initial denaturation (95˚C for 5min), 30 cycles of denaturation (95˚C for 45sec), annealing (55˚C for 5 min), Taq polymerase extension (72˚C for 5 min), and final extension (72˚C for 10 min). The samples were precipitated, and the DNA pellets were re-suspended in sequencing gel-loading buffer [12]. The end-labeled DNA products were resolved on 6% polyacrylamide with 7M urea, DNA sequencing-gels.

## Quantification of CPD yield

The filter membranes were exposed to phosphor storage plates in conjunction with a Typhoon imager (GE Healthcare). Quantifications of gel band signals were done with the ImageQuant software (GE-Healthcare). For the T4-V assay, measurements of CPDs in each strand of Y' sub-telomeres were made as previously described [59]. For the primer extension assay, signals were analyzed by drawing a line tightly around the bands and correcting for background. The density of each band was transferred to an Excel spreadsheet and the frequency (F) of a single photoproduct (or cluster of photoproducts) was measured by quantifying its signal intensity divided by the signal derived from the whole lane. The ratio of signal densities that were measured in the -UV lane was subtracted to correct for signal noise (background). The percent of repair for each photoproduct was plotted over the incubation time, whereby the values measured for 0 hours repair corresponded to 100% damage (or 0% repair), and percent of repair = 100 x [(F(time 0)—F(repair time point)) / F(time 0)] [60].

The raw data for all repair assays included here have been deposited on Mendeley and are accessible via this link https://data.mendeley.com/datasets/j9869tptct; or via this identifier doi: 10.17632/j9869tptct.

## TERRA transcription start-site (TSS) mapping

TSS were mapped by 5' rapid amplification of cDNA ends (5'-RACE). To enrich the TERRA transcripts, total RNA was purified from *sir2Δ* yeast, which have higher levels of TERRA than WT (*SIR2$^+$*) [43]. After purification, the nucleic acids were treated with DNaseI and the total RNA was treated with Ribo-Zero to eliminate rRNA, in the presence of RiboGuard RNase inhibitor. The purified RNA was de-capped with the tobacco acid pyrophosphatase (TAP).

The resulting RNA 5'-end phosphate groups were ligated by the T4 RNA ligase to the RNA oligonucleotide (S2 Table) having the complementary sequence for the forward primer. After purification, the reverse transcription was done with a CA-rich primer targeting TERRA (S2 Table), together with the actin specific primer (S2 Table) that was used as control. The actin TSS was first mapped to assess the quality of both, purified RNA and synthesized cDNA.

Amplification of the cDNA was done by touch-down and nested PCR, using oligo nucleotides complementary to the oligo RNA sequence (forward primer), and the X-element subtelomeric sequences of the 1L, 2R, 3R, 13R, and 15L chromosomes (reverse primers) (S2 Table). Annealing to the subtelomeric sequences allowed a maximum of one mismatch. For the touch-down reaction, 2 μl of the reverse transcription was added to 2.5 μl of 10 μM 5'-RACE forward primer, 2.5 μl of 10 μM oBL293 (or oBL262) reverse primer (S2 Table), 25 μl Phusion master mix, and 18 μl H$_2$O. The touch down-PCR program was the following: 98˚C for 30 sec; (5x) 98˚C for 30 sec and 72˚C for 30 sec; (5x) 98˚C for 30 sec and 70˚C for 30 sec; (25x) 98˚C for 10 sec, 60˚C for 20 sec, 72˚C for 30 sec; 72˚C for 10 min. For the nested PCR, 2 μl of the touch-down PCR was added to 2.5 μl of 10 μM 5'-RACE nested forward primer and 2.5 μl of 10 μM oBL296 (S2 Table), 25 μl 2x Phusion master mix, and 18 μl H$_2$O. The nested PCR program was the following: 98˚C for 30 sec; (35x) 98˚C for 10 sec, 60˚C for 20 sec, 72˚C for 35 sec; 72˚C for 10 min.

Cloning and sequencing of TERRA TSS. PCR products were cloned using the Zero Blunt TOPO PCR kit (Thermo Fisher). Plasmids were isolated from the positive transformants using the Directprep 96 Miniprep Kit (Qiagen), and sequenced. The resulting sequences were blasted against the *S. cerevisiae* S288C genome. The last nucleotide preceding the RNA oligo nucleotide was considered as the TERRA TSS.

## Supporting information

**S1 Table. Yeast strains.**
(TIF)

**S2 Table. Primers.** Y'-element forward (Y'F) and reverse (Y'R) primers were used as [32]P end-labeled single strand DNA probes (Fig 1A). Four X-element primers ('a' to 'd') were used in the Taq polymerase primer extension assay (Fig 1B). Because of high sequence similarity between repeat elements present at chromosome ends, only few unique primers were found to be specific for the X element at telomere 15L. Oligo nucleotides that were used for the 5' RACE are shown.
(TIF)

**S1 Fig. Schema of restriction enzyme-fragment sizes analyzed in the repair assays on Y' elements.** In common lab strains, about 50% of telomeres have one copy of a Y' element telomere proximal and about 30% of those have multiple copies. When two or more Y'-elements occur on a telomere, they always have the same orientation (arrows on top). Although highly homologous, there are two areas of variability amongst Y' elements, one of them of 1.5 kb (large striped box) and a smaller one of about 0.3 kb (small striped box). The presence or absence of the large area is denoted as Y'-long (~ 6.7 kb) and Y'-short (~ 5.2 kb) respectively. Presence or absence of the small area is not annotated but indicated as v1 and v2 in the drawing. Presence or absence of these variable elements causes occurrence of multiple restriction enzyme fragment sizes for the overlap fragment, as indicated. For the yeast used in this study, *HindIII* and *EcoRI* double digestion resulted in three overlap DNA fragments with similar lengths (4.7 kb, 5.0 kb and 4.4 kb). Note that a Y' v2 –Y'-short combination does not exist in the strains analyzed here. The *HindIII* digestion also released a ~ 3.0 kb terminal fragment covering the

telomere, and *Xho*I released a terminal fragment of ~ 1.3 kb. Because of the variable length of the telomeric repeats, these latter terminal fragments appear smeary on gels.
(TIF)

**S2 Fig. NER in the Y'-element of *rad14Δ* yeast.** WT and *rad14Δ* yeast were UV irradiated, and the DNA was prepared for the T4-V assay as described in **Fig 2**. The band signals corresponding to the ~4.7 and ~3.0 kb fragments were measured as described in **Fig 2**, and the resulting means of 2 independent experiments for the *rad14Δ* yeast were plotted against the results obtained for the WT strain (see **Fig 2**). Quantifications were of CPDs for the *HindIII/ EcoRI* group of bands (~4.7 kb), and for the bands with variable lengths (~3.0 kb) as pointed by the brackets. The results show that CPDs in the Y'-element are not repaired in the *rad14Δ* strain.
(TIF)

**S3 Fig. NER in the Y'-element telomeres.** WT and *rad26Δ* yeast were UV irradiated, and the DNA was prepared for the T4-V assay as described in **Fig 2**. The band signals corresponding to the ~4.7, ~3.0 and ~1.3 kb fragments were measured as described in **Fig 2**, and the resulting means of 3 independent experiments ± 1SD were plotted to compare repair in the C- and G-rich strand of WT and *rad126Δ* yeast. *P*-values were calculated using a 2way ANOVA test: (**) $p < 0.01$, (****) $p < 0.0001$.
(TIF)

**S4 Fig. Repair plots of PDs in the Tel15L X-element of WT and *rad26Δ* cells, G-rich strand.** Upper panel: X-element map with the XC- and XR- regions, telomeric repeats (T) and flanking DNA (lines). Approximative DNA sequence positions of primer 'c', *Rsa*I restriction site, PDs that were quantified (black bars) and numbered negatively or positively when present outside or inside of the X element, respectively. Lower panel: repair of PDs (-3 to +17) is plotted as percent of repair over time (hours); WT (diamond, continuous line) and *rad26Δ* (circle, dotted line). Indicated at the top of each plot is the arbitrary number of the PD, with its position on the DNA sequence in parenthesis; both numbers refer to the beginning of the X-element. Means are for 2 independent experiments, and the means ± 1SD are of 3 independent experiments.
(TIF)

**S5 Fig. Repair plots of PDs in the Tel15L X-element of WT and *rad26Δ* cells, G-rich strand.** Upper and lower panels are as described in **S4 Fig**, with the approximative DNA sequence positions of primers 'd', and of the *Hha*I restriction site. Lower panel: repair of PDs (+18 to +27) is plotted as percent of repair over time (hours); WT (diamond, continuous line) and *rad26Δ* (circle, dotted line. Means are for 2 independent experiments, and the means ±1SD are of 3 independent experiments.
(TIF)

**S6 Fig. Repair plots of PDs in the Tel15L X-element of WT and *rad26Δ* cells, C-rich strand.** Upper and lower panels are as described in **S4 Fig**, with the approximative DNA sequence positions of primer 'a' and of the *Hha*I restriction site. Lower panels: PDs (-17 to +10) is plotted as percent of repair over time (hours); WT (diamond, continuous line) and *rad26Δ* (circle, dotted line). Means are for 2 independent experiments, and the means ±1SD are of 3 independent experiments. Grey boxes represent the region of PDs where TC-NER is observed.
(TIF)

**S7 Fig. Repair plots of PDs in the Tel15L X-element of WT and *rad26Δ* cells, C-rich strand.** Upper and lower panels are as described in **S4 Fig**, with the approximative DNA sequence

positions of primers 'a' and 'b', and of the *Hha*I restriction site. Lower panels: repair of PDs (+11 to +36) is plotted as percent of repair over time (hours); WT (diamond, continuous line) and *rad26Δ* (circle, dotted line). Means are for 2 independent experiments, and the means ±1SD are of 3 independent experiments.
(TIF)

**S8 Fig. Repair plots of PDs in the Tel15L X-element of WT and *sir2Δ* cells, G-rich strand.** Upper and lower panels are as described in **S4 Fig,** with the approximative DNA sequence positions of primer 'c' and of the *Rsa*I restriction site. Lower panels: Repair of PDs (-3 to +17) is plotted as percent of repair over time (hours) WT (diamond, black line) and *sir2Δ* (square, grey line). Means are for 2 independent experiments, and the means ±1SD are of 3 independent experiments.
(TIF)

**S9 Fig. Repair plots of PDs in the Tel15L X-element of WT and *sir2Δ* cells, G-rich strand.** Upper and lower panels are as described in **S4 Fig,** with the approximative DNA sequence positions of primer 'd' and of the *Hha*I restriction site. Lower panels: Repair of PDs (+18 to +27) is plotted as percent of repair over time (hours); WT (diamond, black line) and *sir2Δ* (square, grey line). Means are for 2 independent experiments, and the means ±1SD are of 3 independent experiments.
(TIF)

**S10 Fig. Repair plots of PDs in the Tel15L X-element of WT and sir2Δ cells, C-rich strand.** Upper and lower panels are as described in **S4 Fig,** with the approximative DNA sequence positions of primer 'a' and of the *Hha*I restriction site. Lower panels: Repair of PDs (-17 to +11) is plotted as percent of repair over time (hours); WT (diamond, black line) and *sir2Δ* (square, grey line). Means are for 2 independent experiments, and the means ±1SD are of 3 independent experiments.
(TIF)

**S11 Fig. Repair plots of PDs in the Tel15L X-element of WT and *sir2Δ* cells, C-rich strand.** Upper and lower panels are as described in **S4 Fig,** with the approximative DNA sequence positions of primer 'b' and of the *Hha*I restriction site. Lower panels: Repair of PDs (+12 to +37) is plotted as percent of repair over time (hours); WT (diamond, black line) and *sir2Δ* (square, grey line). Means are for 2 independent experiments, and the means ±1SD are of 3 independent experiments.
(TIF)

**S12 Fig. Mapping of TERRA TSS at the 3'-end of the X-element for chromosome 2R, 12R, 13L and 15L.** (A) 5'-RACE. The tobacco acid pyrophosphatase (TAP) was used to remove the 5' caps of purified RNA from *sir2Δ* cells. A defined RNA oligo was ligated to the uncapped RNA and the product was reverse transcribed with a telomeric-repeat reverse primer. The TERRA cDNA was amplified by touch-down and nested PCR reactions, using the forward primer complementary to the RNA oligo sequence and the reverse primers specific to a subset of TERRA molecules. The PCR products were separated by agarose gel electrophoresis (+ TAP) and compared to the products of control samples (- TAP). (B) The PCR products (+TAP) were cloned into sequencing vectors. The sequences were blasted against the yeast genome and TERRA TSS were mapped (see **Fig 7B**). Shown are the number of hits per telomere/TERRA. (C) Schema of the sub-telomeric region with the approximative position of the TERRA TSS at the 3'-end of the X-element for telomere 1L, 2R, 13L,12R and 15L. Upper for the X only telomere and lower for the Y' telomere. 1L TERRA TSS was previously published

[42]; STR: subtelomeric repeated region; Arrow heads: telomeric repeats.
(TIF)

**S13 Fig. Repair plots of PDs in the Tel15L X-element of *sir2Δ* and *sir2Δ rad26Δ* cells, G-rich strand.** Upper and lower panels are as described in **S4 Fig,** with the approximative DNA sequence positions of primer 'c' and of the *Rsa*I restriction site. Lower panels: Repair of PDs (-3 to +17) is plotted as percent of repair over time (hours); *sir2Δ* (square, continuous line) and *sir2Δrad26Δ* (circle, dashed line). Means are for 2 independent experiments, and the means ±1SD are of 3 independent experiments. Grey boxes represent the region of PDs where TC-NER is observed.
(TIF)

**S14 Fig. Repair plots of PDs in the Tel15L X-element of *sir2Δ* and *sir2Δ rad26Δ* cells, G-rich strand.** Upper and lower panels are as described in **S4 Fig**, with the approximative DNA sequence positions of primer 'd' and of the *Hha*I restriction site. Lower panels: Repair of PDs (+18 to +27) is plotted as percent of repair over time (hours); *sir2Δ* (square, continuous line) and *sir2Δrad26Δ* (circle, dashed line). Means are for 2 independent experiments, and the means ±1SD are of 3 independent experiments.
(TIF)

**S15 Fig. Repair plots of PDs in the Tel15L X-element of *sir2Δ* and *sir2Δ rad26Δ* cells, C-rich strand.** Upper and lower panels are as described in **S4 Fig**, with the approximative DNA sequence positions of primer 'a' and of the *Hha*I restriction site. Lower panels: Repair of PDs (-17 to +8) is plotted as percent of repair over time (hours); *sir2Δ* (square, continuous line) and *sir2Δrad26Δ* (circle, dashed line). Means are for 2 independent experiments, and the means ±1SD are of 3 independent experiments.
(TIF)

**S16 Fig. Repair plots of PDs in the Tel15L X-element of *sir2Δ* and *sir2Δ rad26Δ* cells, C-rich strand.** Upper and lower panels are as described in **S4 Fig**, with the approximative DNA sequence positions of primer 'a' and 'b' and of the *Hha*I restriction site. Lower panels: Repair of PDs (+9 to +37) is plotted as percent of repair over time (hours); *sir2Δ* (square, continuous line) and *sir2Δrad26Δ* (circle, dashed line). Means are for 2 independent experiments, and the means ±1SD are of 3 independent experiments. Grey boxes represent the region of PDs where TC-NER is observed.
(TIF)

## Acknowledgments

We thank Hoang Dong Nguyen for his help with the bioinformatics analyses and the members of the Conconi laboratory for useful discussions.

## Author Contributions

**Conceptualization:** Laetitia Guintini, Marco Graf, Brian Luke, Raymund J. Wellinger, Antonio Conconi.

**Data curation:** Audrey Paillé.

**Formal analysis:** Laetitia Guintini, Audrey Paillé, Marco Graf, Raymund J. Wellinger.

**Funding acquisition:** Brian Luke, Raymund J. Wellinger, Antonio Conconi.

**Investigation:** Laetitia Guintini, Audrey Paillé, Marco Graf.

**Methodology:** Laetitia Guintini, Marco Graf, Raymund J. Wellinger, Antonio Conconi.

**Project administration:** Laetitia Guintini, Brian Luke, Raymund J. Wellinger, Antonio Conconi.

**Resources:** Brian Luke, Raymund J. Wellinger, Antonio Conconi.

**Supervision:** Brian Luke, Antonio Conconi.

**Validation:** Marco Graf, Brian Luke, Antonio Conconi.

**Visualization:** Laetitia Guintini, Audrey Paillé, Marco Graf, Antonio Conconi.

**Writing – original draft:** Audrey Paillé, Antonio Conconi.

**Writing – review & editing:** Audrey Paillé, Raymund J. Wellinger, Antonio Conconi.

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
