## [Decision Letter · Decision Letter 0]

8 Nov 2021

Dear Dr Wellinger,

Thank you very much for submitting your Research Article entitled 'Transcription of ncRNAs promotes repair of UV induced DNA lesions in Saccharomyces cerevisiae subtelomeres.' to PLOS Genetics.

The manuscript was fully evaluated at the editorial level and by independent peer reviewers. The reviewers appreciated the attention to an important problem, but raised some substantial concerns about the current manuscript. Based on the reviews, we will not be able to accept this version of the manuscript, but we would be willing to review a much-revised version. We cannot, of course, promise publication at that time.

If you decide to revise the manuscript for further consideration at PLOS Genetics, please aim to resubmit within the next 60 days, unless it will take extra time to address the concerns of the reviewers, in which case we would appreciate an expected resubmission date by email to plosgenetics@plos.org.

[LINK]

We are sorry that we cannot be more positive about your manuscript at this stage. Please do not hesitate to contact us if you have any concerns or questions.

Yours sincerely,

Lorraine S. Symington

Associate Editor

PLOS Genetics

Gregory P. Copenhaver

Editor-in-Chief

PLOS Genetics

Reviewer's Responses to Questions

**Comments to the Authors:**

Reviewer #1: In this elegant study the authors use budding yeast as a model system to uncover a novel role for transcription coupled nucleotide excision repair in removal of UV photoproducts in subtelomeric DNA. A major strength of the study is analysis of photoproduct repair at nucleotide resolution in a strand specific manner. More rapid repair on the C-rich strand, compared to the G-rich strand, in the 3.0 kb Y’ elements is consistent with a greater abundance of Sub-TERRA-XUTs compared to Sub-TERRA-CUTs, and more transcription on the C-rich strand. The manuscript is well written and rigorous in analyzing various sections of the subtelomere and telomeres. Addition statistical analyses comparing wt and mutant strains would strengthen the conclusions. Overall, the study was very well done, and I have a few comments below.

1. In the introduction it would be useful to mention that Rad26 is the yeast homolog of the human CSB gene, and mutations in CSB cause the degenerative disorder Cockayne’s syndrome. I believe this would be of interest to the readership and underscores the relevance for human disease.

2. Figure 2. How do the authors know that they have adding saturating amounts of T4 endo nuclease; amounts sufficient to cleave all the CPDs?

3. The authors note that the formation of photoproducts depends on sequence. Are the sequences in the Y’ and X elements known?

4. The graphs in Figs 2-4 lack statistical comparisons between WT and rad26 null strains. Statistical comparison between C-rich and G-rich strand repair would also strengthen the conclusions, particularly for Figs 2B and 2C.

5. Do the authors have an estimate of how many CPDs were induced on the various elements and fragments?

6. Fig 2. Are both strands transcribed to a similar degree? Are Sub-TERRA-XUTs similar in abundance to Sub-TERRA-CUTs? The authors reference a manuscript that indicate the former is more abundant.

7. Fig 5. Statistical comparison between repair in WT and sir2 null strains would strengthen the conclusions.

8. Treatment with general transcription inhibitors (i.e. thiolutin) are typically used as controls in TCR assay. Have the authors tried transcription inhibitors?

9. Discussion. It is not clear why Sir2 would impair NER. Do the authors have a suggestion? Is there any evidence that Rap1, Rif1 and Sir proteins impair lesion recognition by the Rad4-Rad23 complex in GGR?

10. Discussion. While the manuscript is focused on NER in yeast, it would be interesting to compare results with what is known regarding NER at telomeres in human cell lines, especially since genetic defects in TCR and GGR genes give rise to genetic disorders (XP, Cockayne’s syndrome, TTD).

Reviewer #2: The manuscript by Guintini and colleagues, entitled "Transcription of ncRNAs promotes repair of UV induced DNA lesions in Saccharomyces cerevisiae subtelomeres,” characterizes the potential role of non-coding RNA transcription in sub-telomeric regions of the yeast genome in promoting repair of UV damage via the transcription coupled-nucleotide excision repair (TC-NER) pathway. The authors present evidence that the Rad26-dependent TC-NER pathway promotes repair of UV damage in specific regions of X and Y-elements in yeast telomeres. While the authors have beautiful repair data supporting many of their conclusions, there are number of significant concerns. These concerns are detailed below.

1. In a previous publication (Guintini et al., NAR, 2017), the authors showed that Rad26 contributed to repair of UV damage at Y’ elements at yeast telomeres. Since this finding overlaps to some degree with the current report, it would be important for the authors to distinguish the major advances of the new study, relative to these previous findings.

2. It is common in the field to measure TC-NER in a rad16 mutant strain, in order to avoid potential complications due to differential GG-NER. Since some of the differences in repair attributed to TC-NER are relatively modest, it might be helpful to try some of these experiments in a rad16 mutant background.

3. The high-resolution repair analysis of the Tel-15L telomere (e.g., figures 3, 4,etc) is beautiful. However, the interpretation of the data is limited to some degree by the lack of accurate ncRNA/transcription data for this genomic region. It might be helpful to try to generate this data to assist in the analysis.

4. It would also be really helpful to do a similar high-resolution analysis of the Y’ element, in order to see whether the TERRA transcript corresponds with the regions of high TC-NER in these sub-telomeric regions.

5. The impact of the study could be improved by seeing if the TC-NER activity detected in yeast is also associated with non-coding TERRA expression in human telomeres. This could be analyzed Bioinformatically using published XR-seek data from XPC mutants cells (e.g., Hu et al., 2015).

Reviewer #3: In this study, the authors examine the repair of UV-induced DNA damage at subtelomeric regions in S. cerevisiae. Nucleotide excision repair (NER) is important for the repair of UV-induced damage. There are two types of NER: global genome NER (GGR), and a second type called transcription-coupled NER (TCR) that is specifically active on the transcribed strand of active genes. There are two types of subtelomeres in yeast: one that contains X and Y’ elements, and one that contains only X elements. X elements are known to be transcriptionally silenced, while Y’ elements are not. In addition, the terminal telomeric repeats themselves are transcribed into long non-coding RNA called TERRA. This transcriptional activity at telomeres and subtelomeres may influence the repair of UV-induced damage. The authors use two different assays to study this, one to examine repair at Y’ elements and one to examine repair at an X-only telomere. The manuscript is well written and the quality of the data is, in general, high. However, there are major shortcomings that need to be addressed, as detailed below.

Major points:

The authors talk exclusively about NER, but at the dose used in this study (180 J/m2), several DNA repair pathways are active. For example, homologous recombination mutants and mutants of the post-replicative repair pathway are very sensitive to such a dose. The likely involvement of these pathways dramatically changes the interpretation of all the results. The authors could use an NER mutant in their system to show that all repair is proceeding via NER.

A major part of this study uses sir2∆ to examine the effect of increased transcription on the repair of UV-induced damage. While repair is more efficient in sir2∆, this effect is mostly TCR independent, except for a few clusters affected by deletion of RAD26. Thus, there is very little insight into how deletion of SIR2 actually improves repair, which repair pathways are involved, and whether increased transcription is responsible. Are sir2∆ cells more resistant to UV?

Other points:

Line 169: It is unclear how 4.4, 4.7, and 5.0 kb bands are obtained. Perhaps a diagram could help.

It is mentioned that NER efficiency decreases in a centromere-to-telomere direction, but the data does not convincingly support this assertion conclusion. It is difficult to compare the 4.7, 3.0, and 1.3 kb fragments because the amount of initial damage decreases with decreasing fragment size. Also, as mentioned by the authors, NER efficiency depends on both DNA sequence and length, and these two variables are not controlled for.

In Fig. 2, why are there different banding patterns in 2A and 2D, and also 2B and 2E? It is unclear why deletion of RAD26 would cause such an effect.

In Fig. 3, why are natural pause sites (*) on the G-rich strand different between WT and rad26∆? Similarly, comparing Fig. 5 with Fig. 4, why are the natural pause sites on the C-rich strand different for sir2∆ than WT?

In Fig. S1B, where are PDs 19-21?

Fig. S1A/B maps of PDs do not match that the map in Fig. 3B (note the position of the PDs on either side of the XR/XC boundary).

From the gel shown in Fig. 4A, it appears that repair of PDs -17, -16, and -15 is defective in rad26∆, but the corresponding graphs in Fig. 4 and Fig. S2 say otherwise. Can the authors explain this?

Typo in legends for Fig. 5/6: “The DNA was prepared and ASSED…”

Since the clusters are defined differently for WT (Fig. 3 and 4) than sir2∆ (Fig. 5) (i.e. they do not correspond to the same the regions of DNA), how were the WT and sir2∆ clusters compared in Fig. 5C and 5D?

Fig. 7 is discussed before Fig. 6.

Where is cluster VI for Figure 6A/B?

In the Fig. 6 legend, it should be sir2∆ (square, continuous line) and sir2∆ rad26∆ (circle, dotted line). For all graphs in all figures, there should be legends defining the lines in the figures themselves (not just a description in the figure legends).

Typo on line 302: “extend” should be “extent”.

On lines 307-308, the authors propose that increased TERRA transcription increases TCR of cluster -XIa and cluster -XIIa. If this is indeed the case, and the effect is caused by a single transcript (i.e. TERRA), why would the intervening cluster -XIIb not also be affected?

It is unclear whether Tel1L TERRA transcription can be directly compared with Tel15L. Could the authors map the transcriptional start site for Tel15L using the same approach as Pfeiffer and Lingner?

In Fig. 7, it is unclear what the black and white arrows mean.

Recent work from d’Adda di Fagagna’s lab suggests that telomeric damage-induced long non-coding RNA (tdilncRNA) is important to repair damage at telomeres. Can the authors speculate whether such RNA could be important to repair UV-induced damage at yeast telomeres?

**Have all data underlying the figures and results presented in the manuscript been provided?**

Reviewer #1: Yes

Reviewer #2: Yes

Reviewer #3: Yes

PLOS authors have the option to publish the peer review history of their article (what does this mean?). If published, this will include your full peer review and any attached files.

Reviewer #1: No

Reviewer #2: No

Reviewer #3: No

---

## [Decision Letter · Decision Letter 1]

17 Mar 2022

Dear Dr Wellinger,

Thank you very much for submitting your Research Article entitled 'Transcription of ncRNAs promotes repair of UV induced DNA lesions in Saccharomyces cerevisiae subtelomeres.' to PLOS Genetics.

The manuscript was fully evaluated at the editorial level and by independent peer reviewers. The reviewers were for the most part satisfied with the revisions to the manuscript but raised two minor concerns that need to be addressed.

We therefore ask you to modify the manuscript according to the review recommendations. Your revisions should address the specific points made by each reviewer.

[LINK]

Yours sincerely,

Lorraine S. Symington

Associate Editor

PLOS Genetics

Gregory P. Copenhaver

Editor-in-Chief

PLOS Genetics

Reviewer's Responses to Questions

**Comments to the Authors:**

Reviewer #1: The authors have satisfied all my prior concerns

Reviewer #2: The authors have adequately addressed the concerns and comments mentioned in my previous review.

Reviewer #3: This revised manuscript is a substantial improvement. I am satisfied by their answers. I have only minor comment that remains unresolved.

In the new Figure S1, it is still unclear how the 4.4, 4.7, and 5.0 kb bands are obtained. Looking at the schematic, double digestion with HindIII and EcoRI of either short Y’-short Y’ and long Y’-short Y’ should yield the same HindIII-EcoRI fragment, while short Y’-long Y’ and long Y’-long Y’ should both yield a HindIII-(HindIII) fragment. So assuming complete double digestion, there should only be two bands.

**Have all data underlying the figures and results presented in the manuscript been provided?**

Reviewer #1: Yes

Reviewer #2: **No: **Not clear if underlying numerical data for repair shown in graphs is included in supporting information

Reviewer #3: Yes

PLOS authors have the option to publish the peer review history of their article (what does this mean?). If published, this will include your full peer review and any attached files.

Reviewer #1: No

Reviewer #2: No

Reviewer #3: No

---

## [Editor Report · Decision Letter 2]

25 Mar 2022

Dear Dr Wellinger,

We are pleased to inform you that your manuscript entitled "Transcription of ncRNAs promotes repair of UV induced DNA lesions in Saccharomyces cerevisiae subtelomeres." has been editorially accepted for publication in PLOS Genetics. Congratulations!

Yours sincerely,

Lorraine S. Symington

Associate Editor

PLOS Genetics

Gregory P. Copenhaver

Editor-in-Chief

PLOS Genetics

Comments from the reviewers (if applicable):

**Data Deposition**

http://datadryad.org/submit?journalID=pgenetics&manu=PGENETICS-D-21-01349R2

**Press Queries**

---

## [Editor Report · Acceptance letter]

24 Apr 2022

PGENETICS-D-21-01349R2 

Transcription of ncRNAs promotes repair of UV induced DNA lesions in Saccharomyces cerevisiae subtelomeres. 

Dear Dr Wellinger, 

We are pleased to inform you that your manuscript entitled "Transcription of ncRNAs promotes repair of UV induced DNA lesions in Saccharomyces cerevisiae subtelomeres." has been formally accepted for publication in PLOS Genetics! Your manuscript is now with our production department and you will be notified of the publication date in due course.

With kind regards,

Agnes Pap

PLOS Genetics

On behalf of:
